# Applying the Personal and Social Responsibility Model-Based Program: Differences According to Gender between Basic Psychological Needs, Motivation, Life Satisfaction and Intention to be Physically Active

**DOI:** 10.3390/ijerph16132326

**Published:** 2019-07-01

**Authors:** David Manzano-Sánchez, Alfonso Valero-Valenzuela, Antonio Conde-Sánchez, Ming-Yao Chen

**Affiliations:** 1Department of Physical Activity and Sport, CEI Campus Mare Nostrum, Universidad de Murcia, 30720 Santiago de la Ribera, Spain; 2Department of Statistics and Operational Research, Universidad de Jaén, 23071 Jaén, Spain; 3Department of Sports Information and Communication, University of Aletheia, New Taipei City 25103, Taiwan

**Keywords:** personal autonomy, physical education, gender, social responsibility

## Abstract

The objective of the study was to evaluate the impact of a program based on the Teaching Personal and Social Responsibility (TPSR) on the variables of responsibility, basic psychology needs, motivation, satisfaction with life and the intention to be physically active, as well as the differences of gender. The participants were 85 students (experimental group *n* = 35, 17 girls and control group *n* = 50, 28 girls). The students of the experimental group received the TPSR for 8 months within the physical education subject. The findings indicated an improvement in the experimental group in terms of personal responsibility and in the case of female students, in basic psychological needs and intrinsic motivation. In conclusion, the TPSR program can be integrated into the physical education curriculum in order to improve the personal responsibility of students and fulfill their motivation and satisfaction of basic psychological needs.

## 1. Introduction

Physical education (PE) has being studied as a means to promote sports and their positive values and adherence to sports practice [1]. However, the traditional approach of the current teaching methodologies result in many students justifying their lack of physical activity with bad experiences in PE classes which lead to negative impressions of PE and physical activity in adult life, particularly for female students [2]. Therefore, it is crucial to take a teaching approach that promotes students’ autonomy as well as the improvement of basic psychological needs and motivation [3]. In this perspective, the Teaching Personal and Social Responsibility (TPSR) model-based program, assumes that students need to learn to be responsible for themselves and others in order to socially interact in a suitable way [4,5]. This model-based program is considered as one of the most effective approaches in terms of developing values in the adolescent stage given the positive results it has achieved [6]. Life satisfaction and lower academic stress are strongly related to personal responsibility levels and to academic performance [7], positive personal and social development [8] and a positive impact on students with a tendency to drop out [9]. 

In terms of gender, the studies revealed that levels of personal and social responsibility increased both for male and female students after the TPSR model was applied [10], but differences were significant in boys compared to girls [11]. Therefore, it is important to analyze starting motivation levels given that they might be the result of this stereotyping. Male students generally showed higher satisfaction levels in terms of autonomy, competence and social relatedness needs [12], as well as in intrinsic motivation in relation to female students [12,13]. However, there is a lack of studies relating to the TPSR and its relation with gender on the above variables.

In view of all this, this study assessed the effects of the personal and social responsibility model-based program on responsibility, basic psychological needs satisfaction (BPNS), motivation, life satisfaction and the intention to be physically active to gender.

## 2. Materials and Methods

A quasi-experimental pre-test-post-test non-equivalent group design was used [14]. The authors selected a probability and intentional sample of students and carried out our pre and post- assessment measurements either side of an intervention program based on the promotion of responsibility. 

### 2.1. Participants

Participating secondary and high schools had similar socio-demographic profiles, and they were selected by accessibility and convenience. The necessary authorizations were obtained from the Ethics Committee of the researchers’ university and the schools’ principals. Informed written consent was also obtained from the parents. A total of 85 participants were involved in the study once exclusion criteria were applied: (a) Completing pre- and post-tests, and (b) answer at least 90% of the test items without double counting answers. None of the participants had any previous experience with TPSR. After calculating the Mahalanobis distance to eliminate atypical subjects, two groups were formed made up of 18 boys and 17 girls for the experimental group (*n* = 35) and 28 girls and 22 boys for the control group (*n* = 50). The participant’s age ranged from 14 to 18 years with a mean age of 16.22 (SD = 0.41). 

### 2.2. Instruments

This study used a closed question questionnaire based on a series of scales to assess the study’s main aims. The questionnaire was divided into two parts; the first section was for socio-demographic variables and included questions on gender and date of birth. The second section included the different questionnaires used in the study:

Motivation: Cuestionario para ver la motivación en clases de educación física CMEF (Questionnaire to assess motivation in the physical education class) [15]. Its aim was to assess participants’ intrinsic and extrinsic motivation and amotivation with a 1–5 Likert-type scale including a total of 20 items. The questionnaire started with the following sentence: “I participate in PE classes…”. Pre and post-test reliability values were α = 0.74 and α = 0.78 respectively. The sub-scales’ pre-test and post-test values were α = 0.81 and α = 0.84 respectively, for intrinsic motivation, α = 0.80 and α = 0.80 for extrinsic motivation and α = 0.85 and α = 0.73 for amotivation. This study also used the self-determination index (SDI) to measure all three variables [16]. 

Responsibility: Personal and social responsibility questionnaire (PSRQ) was translated into Spanish [17], and a 1–5 Likert-type scale including a total of 14 items was used to assess participants’ responsibility. The instructions were presented at the beginning of the questionnaire along with the following statement: “It is normal to behave well at times and badly at other times. We are interested in finding out how you normally behave in PE classes. There are no correct or incorrect answers. Please answer the following questions choosing the option which bests represents your behavior”. Pre and post-test reliability values were α = 0.90 and α = 0.88, respectively, after the elimination of one personal responsibility item which showed a low internal consistency value – item 14 – “I do not set any goals for myself”. The sub-scales showed pre- and post- test values of α = 0.84 and α = 0.88 respectively for personal responsibility and α = 0.84 and α = 0.83 for social responsibility.

Basic Psychological Needs: Psychological need satisfaction in exercise scale (PNSE) was validated for the Spanish education context [18]. A 1–5 Likert-type scale including a total of 12 items, was used to assess participants’ basic psychological needs satisfaction in PE. The questionnaire started with the following stem: “In my PE classes….” This questionnaire showed total pre and post-test consistency values of α = 0.86 and α = 0.87 respectively. Pre and post-test values for the sub-scales were α = 0.76 and α = 0.72 for autonomy, α = 0.85 and α = 0.81 for relatedness and α = 0.79 and α = 0.80 for competence. In relation to this psychological construct, the scale known as Basic Psychological Needs Scale [19] was created to analyze all three basic psychological needs as a single factor.

Intention to be physically active: Cuestionario para comprobar la intención de ser físicamente activos (MIPA) (Questionnaire to assess the intention to be physically active) by Moreno, Moreno and Cervelló [20] was used to assess the participants’ intention to be physically active. A 1–5 Likert-type scale including 5 items was used whose total score revealed level of intention to be physically active, with items such as “I am interested in acquiring good physical shape”. Pre and post-test values were α = 0.95 and α = 0.85.

Satisfaction with Life: Satisfaction with life scale [21], validated for the Spanish context by Atienza, Balaguer and García-Merita [22], was used to assess participants’ satisfaction with their life. A 1–7 Likert-type scale including 5 items, whose total score revealed level of satisfaction with life. All items required specifying level of agreement with each statement. The questionnaire showed pre- and post-test average internal consistency values of α = 0.68 and α = 0.72 respectively.

Other documents included were an informed consent signed by the students and their parents who were minors and a cover letter sent to the school and a report by the Universidad de Murcia Ethics Committee.

### 2.3. Procedure

The intervention program was conducted during 8 months. The content selection was in accordance to the current education law and the same learning units were conducted on both research groups. Prior and at the end of the intervention, the participating groups were administered the questionnaire described earlier, in a quiet environment, for 30 minutes. The participants were encouraged to be sincere in their answers.

### 2.4. Training Meetings

After obtaining approval from the high school and the teacher, the study started. The correct implementation of any program requires specific teacher training [23]. The teacher was trained in the TPSR in a two-phase approach: (1) 5-hour course on TPSR theory and practice; they were explained how to design classroom climates based on the model, and they were provided with global and specific strategies for the development of responsibility in PE; and (2) continuous training; throughout the implementation of the program, the main researcher met with the teacher in three-week cycles: Week-1, the teacher sent to the main researcher the sessions they had created to the first level, and he assessed them and provided feedback to the teacher; week-2, the sessions were implemented in PE, and one of the monthly sessions was filmed and assessed by the research team (at the beginning, every two weeks); week-3, a meeting was held between the teacher and the main researcher; an individual report on the interventions was presented, as well as feedback and suggestions for improvement regarding model implementation. The goal was to develop a class climate to promote responsibility through the application of the TPSR. 

### 2.5. TSPR Intervention Program

Each session format followed Hellison’s [24] five-part proposal: (1) Relational time, teachers interacted with their students to create bonds; (2) awareness talk, teachers tried to put responsibility into practice introducing the level to be worked during the session; (3) physical activity plan, the level selected for the session was embedded in all the tasks; (4) group meeting, at the end of each session, the teacher and students shared their perceptions regarding responsibility in class; and (5) reflection time, students self-evaluated their responsibility. The teachers used general strategies to implement the TPSR (e.g., being an example of respect, setting expectations, providing opportunities for success) and specific ones (e.g., redefining success, personal work plans, responsibility for students of other groups). Likewise, strategies were also used to solve individual conflicts (e.g., progressive separation from the group) and collective ones (e.g., accordion principle), fully integrating TPSR in their PE classes [24].

### 2.6. Session and Intervention Structure

The sessions were structured in four sections following Hellison [25]: (1) Awareness talk; (2) responsibility in action; (3) group meetings for reflection; and (4) self-assessment of responsibility. They were carried out following a progression through the five levels of the model and using different and level-specific strategies to solve problems and applying rules of coexistence: (1) One level was respect for other people’s rights and feelings. The teacher attempted to create a suitable atmosphere based on respect for the other classmates, for the material and for the teacher and on self-control; then there was a level of participation and effort; (2) here, the aim was for students to participate in the activities proposed with effort and persistence in the face of adversity; (3) next, a level of personal autonomy, the aim was to promote students’ autonomy and capacity to adjust their own behavior and make decisions; (4) level of helping others and leadership, the objective being to achieve empathetic and group leadership relationships and for students to be able to help without expecting a reward and focusing on those really in need of help; (5) finally, there was an out-of-context level, which consisted of applying the values learned in the previous levels in all contexts of life.

The so called conflict management strategies were also applied. Following the revision carried out [8], the authors used one example of each of these strategies for individual conflicts (five clean days rule, relevant conflict behavior is noted and the student has to avoid it for five days) and for group conflict (the talking bench strategy, with three students designated to settle disputes). 

The teaching units for the second quarter were then defined (basketball, climbing and jump rope) and the sessions, the recordings (explained below) and the tests were scheduled. 

### 2.7. Observations

As Hastie and Casey described [26] to establish fidelity of a model’s implementation, it is necessary to obtain “(a) a rich description of the curricular elements of the unit, (b) a detailed validation of model implementation, and (c) a detailed description of the program context”. Parts A and B have been completed in previous sections of the article.

The analysis was carried out with the tool for assessing responsibility-based education (TARE) [27], which has proved to be a highly reliable and appropriate tool in PE [28]. Thus, the observer noted whether or not the TARE categories were applied during the classes with satisfactory results [24], i.e. over 80% of the total of the items listed for each of the sessions. Following the session analysis process, the teacher received feedback so that he could modify different aspects as required. 

Therefore, a combination of different strategies (i.e., training seminars, video analysis, feedback loops and continued doubt resolution) was used to provide correct guidance and support to the teacher prior to and throughout the whole research project [29].

Before each of the sessions, the PE teacher briefed the researcher on the session to check that it followed TPSR principles. A total of 11 sessions (one of every four-six sessions and the first intervention session) was analysed by external observers (distributed in observation periods of 5 minutes). The teacher’s behaviour was assessed using the TARE instrument. The questionnaire had to be answered on a 1-5 Likert-type scale from, 1 never, to, 5 always. The observational analysis followed the sequence established by Wright and Craig (2011). Total agreement *(TA)* was calculated using the formula: Number of total agreements *(NTA)* divided by agreements *(A)* plus disagreements (D) *(TA = NTA / A + D)* [30]. 

### 2.8. Data Analysis

The analysis of the data was carried out with IBM SPSS 22.0 (SPSS Inc. Chicago, IL, USA). Assessing both its pre- and post-test internal consistency first validated the instrument. Cronbach’s alpha was used to measure reliability. Following that, the research carried out data exploratory analysis using box-whisker plots and descriptive measurements. In view of potential gender-based differences, the inferential analysis carried out was taken into account. 

A first analysis was a repeated measure MANOVA on the ten variables obtained with the different questionnaires; the intra-subject factor was, test (with two levels, pre-test and post-test), whereas group (with two levels, control and experimental) and gender (two levels, male and female) were the inter-subject factors. An analysis of residuals revealed rejection of the normality hypothesis and thus this analysis was ruled out in favor of nonparametric tests. Both procedures revealed very similar results. The MANOVA results are not included here for the sake of brevity.

The normality hypothesis was tested with the Lilliefors and Shapiro-Wilk tests. In terms of nonparametric tests, the sign and Wilcoxon signed-rank tests were used to compare pre- and post-test variables. The Mann–Whitney test was used to compare the variables between the control and experimental groups. This comparison was carried out both pre- and post-test. 

The results were compared with the rule of thumb for effect size suggested by Cohen [31], the effect sizes of 0.10, 0.30, and 0.50 are considered to be small, medium and large, respectively. The analysis included, apart from the ten variables previously described, the self-determination index (SDI) and the basic psychological needs scale (BPNS) to sum up motivation factors and psychological needs respectively.

## 3. Results

The normality tests carried out on MANOVA residuals rejected the normality hypothesis for most of the variables. The research only found p-values over 0.05 for intrinsic motivation and autonomy in the pre-test, life satisfaction in the post-test and extrinsic motivation in both (pre- and post-). Therefore, parametric procedures were ruled out. 

### Inference Results

Pre and post-test means and standard deviations of all the variables differentiated by group and gender are shown in Table 1, which also includes the *p*-values obtained with the different nonparametric comparative tests. 

A comparison of the pre-test variables of the two groups (control and experimental) with the Mann–Whitney *U* test did not reveal significant differences, except for personal responsibility in female students (*p*-value = 0.037), which was higher in the control group in relation to the experimental group. 

It can also be seen that for females, intrinsic motivation was higher in the control group (*p*-value = 0.077) and extrinsic motivation higher in the experimental one (*p*-value = 0.067), though without, in either case, reaching a 5% significance level. This suggests that, in general, the groups were fairly homogenous with regard to the observed variable prior to beginning the social and personal responsibility program. The differences have been taken into account for examining how variables changed between pre-test and post- test.

The bulk of the variables improved throughout the process in both groups and for both genders in terms of median values. The statistically significant differences were observed in the following variables:Intrinsic motivation for female students and the experimental group (*p*-value = 0.019 after Wilcoxon signed-rank test). As a consequence of this increase, the control and experimental tests were levelled post-test. As mentioned before, there were slight pre-test differences in favor of the control group.Extrinsic motivation for female students and the control group (*p*-value = 0.045 after Wilcoxon signed-rank test). In spite of this post-test increase, significant differences were observed in favor of the experimental group (*p*-value = 0.026, Mann–Whitney U test), a difference already observed pre-test.Autonomy for female students and the control group (*p*-value = 0.031 after sign test and *p*-value = 0.042 after Wilcoxon signed-rank test).Competence for female students in both groups (*p*-value = 0.008 and *p*-value = 0.021 after Wilcoxon signed-rank test).Personal responsibility for female students in the experimental group (*p*-value = 0.025 after Wilcoxon signed-rank test). In terms of male students, a significant fall was observed in the control group (*p*-value = 0.019 after the sign test), whereas the experimental group revealed a significant increase (*p*-value = 0.025 after Wilcoxon signed-rank test), with significant post-test differences in favor of the experimental group (*p*-value = 0.012).Social responsibility for female students in the control group (*p*-value = 0.029 after Wilcoxon signed-rank test). No significant post-test differences were observed between the two groups.Basic psychological needs for female students in both groups (*p*-value = 0.031 and *p*-value = 0.020 after Wilcoxon signed-rank test). No significant post-test differences were observed between the two groups.

Furthermore, and though not reaching 5% significance, a series of variables showed p-values which were close to that level: Autonomy (*p*-value = 0.062), relatedness (*p*-value = 0.076) and SDI (*p*-value = 0.051) for female students and the experimental group. Bearing in mind that the size samples were small, the results can be considered to be significant. Moreover, the effect sizes were 0.320 for autonomy, 0.304 for relatedness, and 0.344 for SDI, confirming the differences should be taken into account. Additionally, for those cases with statistically significant differences, the effect sizes generally were medium.

## 4. Discussion

The purpose of this study was to analyze the implementation of the personal and social responsibility model-based program in the context of a high school, specifically with students from secondary and high school (from 14 to 18 years), in order to assess its effects on responsibility, basic psychological needs, motivation, life satisfaction, intention to be physically active and check the differences according to the gender.

Overall, the model-based program achieved an increase in post-test levels of personal responsibility in boys and girls in the experimental group. This result is in line with numerous studies highlighting improvement in responsibility with the application of this model [10,32]. 

This study has revealed the appropriateness of gender differentiation when it comes to comparing results. In terms of the differences between the control and experimental group, male students improved their levels of personal responsibility in the experimental group. There were differences in line with Sánchez-Alcaraz et al. [11] who found the improvement of personal responsibility in the variables analyzed was observed largely in male students, though differences in female students were also found. There were no significant differences found for the rest of the variables, expect for extrinsic motivation in female students with higher values in the experimental group than the control group. 

Regarding satisfaction of basic psychological needs, there were significant differences observed through time for girls in both groups in the BPNS and the competence variable, in line with and Amado et al. [33] who found higher increases for female students. 

In terms of behavior management, pre and post-test differences were observed in intrinsic motivation where female students improved in the experimental group. This is against the results in Sevil et al. [34]. They observed an increase in intrinsic motivation and basic psychological needs in the experimental group and for both genders, especially in boys after applying a program with a control group and an experimental group to promote autonomy and motivation strategies (TARGET areas) through a body expression teaching unit.

Our results tend to be in line with Menéndez and Fernández [35] who found a causal relationship between basic psychological needs and intrinsic motivation. Baena and Granero [36] and Catalán et al. [37] undertook further research and explored the relationships between basic psychological needs and intrinsic motivation. This study also suggested being physically active in the future. 

The gender differences might be a result of female students starting with lower initial levels of responsibility, basic psychological needs and motivation. Lauderdale [30], Granero et al. [38], Nateras and Mendo [39] and Smith [12] provided a similar perspective with their analysis of the characteristics of high school students in the PE class. 

## 5. Conclusions

The TPSR generally improves personal responsibility levels and results are even better in the case of female students, who show higher intrinsic motivation levels in the PE class over time. Improvement was also observed in both groups in terms of the BPNS. However, longer interventions would be required in order to see results in the rest of the variables. 

The main limitations were the small number of PE class hours available to apply the TPSR program and the small size of the sample. Another limitation was related to the teacher. There was no assessment of the teaching methodology that the teacher was using prior to the training process, which might have included some of the strategies used in the TPSR and would have thus mitigated some of the expected effects on the participants. Furthermore, the teaching sessions of the control group were not monitored throughout the research. 

Future lines of research should consider the possibility of applying the TPSR program to other school students so that it can be implemented for a longer period of time. It would also be interesting to apply other variables to the study, including different age ranges and education stages.

## Figures and Tables

**Table 1 ijerph-16-02326-t001:** Pre-and post-test differences according to gender and group.

		PRE-TEST		POST-TEST		Pre-Post Test Differences					
		Male	Female	Male	Female	Male			Female		
						Sign	Wilcoxon		Sign	Wilcoxon	
		Mean (S.D.)	Mean (S.D.)	Mean (S.D.)	Mean (S.D.)	*p*-Value	*p*-Value	Size Effect	*p*-Value	*p*-Value	Size Effect
Intrinsic motivation	Control	4.17 (0.50)	3.89 (0.86)	4.18 (0.51)	4.01 (0.78)	0.629	0.610	−0.077	1	0.365	−0.121
	Experimental	4.26 (0.66)	3.54 (0.75)	4.28 (0.55)	3.81 (0.76)	1	0.964	−0.008	0.092	0.019*	−0.401
	Mann-Whitney (*p*-value)	0.366	0.077	0.581	0.326						
	Size Effect	−0.149	0.263	−0.092	0.146						
Extrinsic motivation	Control	3.13 (0.73)	2.90 (0.50)	3.27 (0.68)	3.07 (0.45)	1	0.592	−0.081	0.307	0.045*	−0.268
	Experimental	3.23 (0.63)	3.28 (0.67)	3.44 (0.74)	3.49 (0.68)	0.332	0.184	−0.221	0.332	0.155	−0.244
	Mann-Whitney (p-value)	0.737	0.067	0.527	0.026*						
	Size Effect	−0.054	−0.273	−0.101	−0.332						
Demotivation	Control	1.72 (1.09)	1.71 (1.07)	1.71 (1.07)	1.69 (1.03)	0.688	0.666	−0.065	1	0.751	0.042
	Experimental	1.57 (0.89)	1.76 (0.83)	1.59 (0.88)	1.75 (0.80)	0.289	0.436	−0.177	1	0.629	0.083
	Mann-Whitney (*p*-value)	0.840	0.373	0.882	0.407						
	Size Effect	0.037	−0.133	0.025	−0.124						
Autonomy	Control	3.42 (0.72)	3.06 (0.79)	3.62 (0.77)	3.36 (0.84)	0.359	0.440	−0.138	0.031*	0.042*	−0.288
	Experimental	3.54 (0.89)	2.90 (0.90)	3.46 (0.99)	3.25 (0.62)	1	0.949	0.092	0.092	0.062	−0.320
	Mann-Whitney (*p*-value)	0.737	0.523	0.737	0.435						
	Size Effect	−0.054	0.095	0.054	0.116						
Competence	Control	4.03 (0.71)	3.57 (0.82)	4.14 (0.58)	3.86 (0.82)	0.824	0.523	−0.096	0.093	0.008**	−0.353
	Experimental	4.10 (0.52)	3.32 (0.81)	4.28 (0.65)	3.72 (0.85)	0.302	0.241	−0.195	0.180	0.021*	0.396
	Mann-Whitney (*p*-value)	0.946	0.383	0.299	0.596						
	Size Effect	−0.011	0.130	−0.167	0.079						
Relatedness	Control	3.83 (0.90)	3.81 (0.83)	4.02 (0.64)	3.79 (0.84)	0.454	0.227	−0.182	0.405	0.702	0.111
	Experimental	4.18 (0.72)	3.72 (0.92)	4.17 (0.81)	4.09 (0.55)	1	0.888	0.024	0.344	0.076	−0.304
	Mann-Whitney (*p*-value)	0.274	0.887	0.396	0.219						
	Size Effect	−0.179	0.021	−0.137	−0.183						
Personal responsibility	Control	4.23 (0.66)	4.36 (0.56)	4.14 (0.56)	4.44 (0.49)	0.019*	0.114	0.346	0.824	0.522	−0.086
	Experimental	4.23 (0.50)	3.79 (0.93)	4.57 (0.47)	4.20 (0.61)	0.057	0.025*	−0.373	0.180	0.025*	−0.386
	Mann-Whitney (*p*-value)	0.925	0.037*	0.012*	0.194						
	Size Effect	0.017	0.311	−0.399	0.194						
Social responsibility	Control	4.24 (0.65)	4.35 (0.45)	4.47 (0.46)	4.49 (0.42)	0.096	0.096	−0.251	0.134	0.029*	−0.292
	Experimental	4.37 (0.59)	4.34 (0.80)	4.53 (0.70)	4.56 (0.38)	0.388	0.344	−0.158	0.388	0.252	−0.197
	Mann-Whitney (*p*-value)	0.581	0.403	0.366	0.587						
	Size Effect	−i.092	−0.125	−1.147	−1.081						
MIPA	Control	4.52 (0.59)	3.69 (1.01)	4.36 (0.83)	3.84 (0.99)	0.804	0.622	0.074	1	0.452	−0.101
	Experimental	4.14 (0.85)	3.64 (0.93)	4.11 (0.60)	3.69 (0.84)	0.607	0.668	0.086	1	0.925	−0.016
	Mann-Whitney (*p*-value)	0.190	0.805	0.125	0.488						
	Size Effect	0.215	0.037	0.248	0.104						
SWLS	Control	5.30 (0.71)	5.21 (0.98)	5.10 (0.90)	5.17 (0.90)	0.167	0.267	0.207	0.664	0.431	−0.089
	Experimental	5.44 (0.90)	5.00 (1.25)	5.57 (0.86)	5.18 (0.90)	0.791	0.571	−0.095	1	0.586	−0.093
	Mann-Whitney (*p*-value)	0.697	0.796	0.199	0.981						
	Size Effect	−0.065	0.039	−0.205	−0.004						
SDI	Control	6.23 (2.85)	6.02 (3.78)	6.46 (2.50)	6.30 (3.55)	0.503	0.360	−0.149	0.845	0.703	−0.026
	Experimental	7.15 (3.64)	4.70 (3.11)	7.01 (2.69)	5.44 (3.01)	0.804	0.698	0.065	0.180	0.051	−0.334
	Mann-Whitney (*p*-value)	0.251	0.160	0.510	0.303						
	Size Effect	−0.185	0.209	−0.105	0.154						
BPNS	Control	3.76 (0.61)	3.48 (0.60)	3.93 (0.58)	3.67 (0.69)	0.523	0.322	−0.138	0.186	0.031*	−0.287
	Experimental	3.94 (0.57)	3.32 (0.74)	3.97 (0.65)	3.69 (0.56)	0.332	0.619	−0.162	0.210	0.020*	−0.399
	Mann-Whitney (*p*-value)	0.427	0.325	0.819	0.944						
	Size Effect	−0.129	0.148	−0.039	−0.010						

Legend: * *p* = < 0.05; ** *p* = < 0.01; S.D. = Standard Deviation; MIPA = Intention to be physically active; SWLS = Satisfaction with life scale; SDI = Self determination index; BPNS = Basic psychological needs

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
