# Peer review of "Applying the Personal and Social Responsibility Model-Based Program: Differences According to Gender between Basic Psychological Needs, Motivation, Life Satisfaction and Intention to be Physically Active"

_ijerph, 2019, doi:10.3390/ijerph16132326_

Round 1

Reviewer 1 Report

The authors study the link between the participation of a program based on Personal and Social Responsability and students' attitudes. The authors use data from an experimental study.

This is an interesting paper and deserves publication in International Journal of Environmental Research and Public Health, if the author can make the following improvements / clarifications:

Line 70-110: No definition of the instruments used is given. The author presents the surveys used to measure the instruments but don't give the meaning of the instruments. These definitions are necessary for a good understanding of the article for a non expert reader.

The author speaks of impact of the program whereas it would be better to talk about links

Without error, no information on the year and on the period of the experiment are given.

Line 211-222 + Line 312-314: copy paste of editing rules.

Line 63 and 67: Warning repetition

The title of Table 1 should be simplified (too long 2 sentences)

The formatting of Table 1 should be improved.

For the non specialist reader, some acronyms that appear in the text should be redefined. For example, BNPS at line 296.

Author Response

Dear Sir/Madam:

Thank you very much for reviewing our manuscript. We greatly appreciate the reviewers for their complimentary comments and suggestions. We have revised the manuscript accordingly. Please find attached a point-by-point response to reviewer’s concerns. We hope that you find our responses satisfactory and that the manuscript is now acceptable for publication.

Sincerely,

Ming-Yao Chen

Department of Sports Information & Communication, Aletheia University.

Email: mingychen@hotmail.com

Reviewer 2 Report

The manuscript is lots of effort to make this happen. The words need to match the effort of the study. At the present, there are too many errors to trust/understand the results. The manuscript needs full cleaning so it can be read again.

Comments

Line 16, add the...based on the Personal and Social Responsibility (TPSR)

Line 29, take out the... Physical education...

Overall, comment on 'the' - you have too many of them.

Line 39, take out The... it is just Life satisfaction and ....

Line 44, what differences are significant in boys?

Lines 45-46, I do not understand your point.

Line 53, To this aim is not the right phrase. I would just continue the sentence ...was used as we selected...

Lines 70-71, ...to assess the study's main aims.

Line 72, I don't understand what questionnaire or set of questions you are referring to with your alpha values.

Line 75/83, I do not understand how you are choosing words to capitalize.

Line 107, and not &

Line 121, the formation started does not sound correct. Maybe the study started.

Lines 127-130, I do not see the need for the -. Just week 1, week 2, etc.

Line 167, the comma should be after implementation

Line 172, Physical education does not need to be capitalized.

Lines 211-222 are from the template.

Line 191-192, I would merge those as one paragraph.

The tables need to be landscape or a smaller font so one can read them.

Line 239 down to the discussion, you stopped using a paragraph indention.

Line 316/328, you are spelling out the model when before you have used TPSR.

Line 321, I still see no need for physical education to be capitalized.

Author Response

(The authors gave the same response as above.)

Round 2

Reviewer 2 Report

Hello, I appreciate your efforts. I need help understanding your effect size values in your table. I see p values that are less than .05 with small effect sizes, and then p values with greater than .05 with larger effect sizes. They do not seem to match up. I know they are not a perfect match. It just seems I should be able to go, sure an effect size > in magnitude of .xx is probably always with a p , .05. 

Perhaps I do not understand all of the effect sizes? What are the exact comparisons being made?

Again, your manuscript contains stock sentences from the publisher. Lines 207-212 are stock sentences, not your sentences.

Line 261-262 is where your effect sizes should tell us the results might not reach conventional statistical significance, but the effect sizes were.... What were they?

Author Response

Hello, I appreciate your efforts. I need help understanding your effect size values in your table. I see p values that are less than .05 with small effect sizes, and then p values with greater than .05 with larger effect sizes. They do not seem to match up. I know they are not a perfect match. It just seems I should be able to go, sure an effect size > in magnitude of .xx is probably always with a p , .05. 

Perhaps I do not understand all of the effect sizes? What are the exact comparisons being made?

We really thank you all your interest for understanding what we have done and make the manuscript more comprehensible. We have been checking all the values obtained, and in those cases where there are significant differences, the effect size is generally above 0.3, meanwhile for those cases with no significant differences, the effect size is smaller. However, it is true that there is a mistake. In particular, when we look for differences between pre and post test for males in the personal responsibility variable, there is an effect of -.0373, and should be -.373 (row 14, column 9 of the table). We have highlighted with red colour the amendment.

Again, your manuscript contains stock sentences from the publisher. Lines 207-212 are stock sentences, not your sentences.

Thank you very much for these appreciations. Honestly, we were not aware of this mistake. The two sentences have been deleted.

Line 261-262 is where your effect sizes should tell us the results might not reach conventional statistical significance, but the effect sizes were.... What were they?

We totally agree with the reviewer. It has been included a new sentence (lines 262-263) describing these values.

Finally, we have amended the rule of thumb values for effect size suggested by Cohen for nonparametric tests (line 204).

Round 3

Reviewer 2 Report

I appreciate all of your efforts.